# Inhibitory Effects of Eriodictyol-7-*O*-*β*-d-glucuronide and 5,7-Dihydroxy-4-chromene Isolated from *Chrysanthemum zawadskii* var. *latilobum* in FcεRI-Mediated Human Basophilic KU812F Cell Activation

**DOI:** 10.3390/molecules25040994

**Published:** 2020-02-23

**Authors:** Mina Lee, Sun-Yup Shim

**Affiliations:** 1College of Pharmacy, Sunchon National University, 255 Jungangno, Suncheon-si, Jeonnam 57922, Korea; minalee@sunchon.ac.kr; 2Department of Aqualife Science, College of Fisheries and Ocean Sciences, Chonnam National University, 50 Daehak-Ro, Yeosu, Jeonnam 59626, Korea

**Keywords:** *Chrysanthemum zawadskii* var. *latilobum*, eriodictyol-7-*O*-*β*-d-glucuronide, 5,7-dihydroxy-4-chromene, FcεRI, histamine, extracellular regulated kinases 1/2

## Abstract

*Chrysanthemum zawadskii* var. *latilobum* (CZL) has been used in Eastern medicine for the treatment of various diseases, such as pneumonia, bronchitis, cough, the common cold, pharyngitis, bladder-related disorders, gastroenteric disorders, and hypertension. In the present study, we isolated two strong antiallergic compounds from CZL, namely, eriodictyol-7-*O*-*β*-d-glucuronide (EDG) and 5,7-dihydroxy-4-chromene (DC), and investigated their antiallergic effects in FcεRI-mediated human basophilic KU812F cells. EDG and DC downregulated the protein and messenger RNA (mRNA) expression of FcεRI on the cell surface. Moreover, Western blotting analysis showed that EDG and DC inhibited the expression of protein tyrosine kinases such as Syk and Lyn, and extracellular-regulated kinases (ERK) 1/2. These results suggested that EDG and DC, antiallergic constituents of CZL, are potential therapeutic candidates for protection against and for the treatment of allergic disorders.

## 1. Introduction

Allergic disorders are increasing rapidly in recent years and are important public-health problems that can affect the quality of life and can create socioeconomic burdens [1]. Antihistamines, anti-inflammatory drugs, and immunosuppressive agents are common drugs currently used to treat and prevent allergic disorders. However, these medications can exert lethal side effects such as cardiotoxic action [2]. Secretion of FcεRI-mediated inflammatory mediators from activated basophils and mast cells is a special feature of immediate hypersensitivity [3,4]. The receptor’s activation cascade is initiated by aggregation of cell-surface FcεRI attached to allergen-specific IgE antibody through the binding of a multivalent allergen [5,6]. Mast cells and basophils play an important role as effector and regulator cells in IgE-mediated allergic diseases owing to their wide distribution, and a remarkable feature of FcεRI expression on these cells has been identified [7]. FcεRI consists of an α-chain that binds to the Fc portion, and signal-transducing β- and γ-chains that contain immunotyrosine-based activation motifs in their cytoplasmic domains [8,9]. The crosslinking of FcεRI molecules attached to allergen-specific IgE antibody through the binding of a multivalent allergen results in the activation of downstream factors related to protein tyrosine kinases (PTK) such as Syk and Lyn, and mitogen-activated protein kinases (MAPKs) such as extracellular-regulated kinases (ERK) 1/2, p38 protein kinase, and c-jun N terminal kinase [10,11]. Aggregation is the major stimulus for effector cells, as it triggers degranulation and calcium influx, resulting in the secretion of various chemical mediators such as histamine, prostaglandins, and leukotrienes from activated effector cells, which causes allergic responses such as asthma, atopic dermatitis, and allergic rhinitis [3,4,11]. The downregulation of FcεRI expression in effector cells may lead to the attenuation of IgE-mediated allergic reaction. The signaling pathways of MAPK and NF-kB are predominant cascades that are closely related to allergic responses [12].

*Chrysanthemum zawadskii* var. *latilobum* (CZL) is a perennial herb of the Compositae family, which is known as “gujeolcho” in Korea and it has been used as traditional medicine for the treatment of various diseases. This plant has various pharmacological properties, including anticancer, oxidative, inflammatory, and hepatoprotective effects [13,14]. It is rich in flavonoids and contains linarin and acacetin, which exert protective effects against cancer and inflammation [15,16,17,18,19]. However, the effects of eriodictyol-7-*O*-*β*-d-glucuronide (EDG) and 5,7-dihydroxy-4-chromene (DC) isolated from CZL on allergic reactions have not been examined. To search for phytochemicals with antiallergic potential, we investigated the in vitro antiallergic activities of EDG and DC isolated from CZL in anti-FcεRI antibody (CRA-1)-induced human basophilic KU812F cells.

## 2. Results

### 2.1. EDG and DC Isolation from CZL

The structures of EDG and DC (Figure 1) isolated from methanol (MeOH) extract of CZL were confirmed as compounds of flavanone glucoside and chromene series on the basis of a comparison with ^1^H- and ^13^C NMR spectra data from the literature [20].

### 2.2. EDG and DC Cytotoxicity

In order to ensure the noncytotoxic concentrations of EDG and DC in KU812F cells, a 3-(4,5-dimethylthiazol-2-yl)-5-(3-carboxymethoxyphenyl)-2-(4-sulfophenyl)-2H-tetrazolium (MTS) assay was performed at 10, 25, 50, and 100 mM of EDG and DC in KU812F cells. EDG and DC did not exhibit any cytotoxic effect on KU812F cells at concentrations of up to 100 µM in comparison with control cells after treatment for 24 h (Figure 2). Subsequent experiments were performed with EDG and DC at concentrations of up to 100 µM.

### 2.3. EDG and DC Effects on FcεRI Expression

FcεRI expression on the cell surface of EDG-treated cells was observed to be reduced from 30.0% to 25.5%, 21.0%, 19.4%, and 16.5% by the treatment of EDG at a level of 10, 25, 50, and 100 μM, respectively (Figure 3A). FcεRI expression in DC-treated cells was observed to be reduced from 29.8% to 25.7%, 22.4%, 20.5%, and 18.1% by treatment of DC at concentrations of 10, 25, 50, and 100 μM, respectively (Figure 3A). To further study the downregulation of these compounds on FcεRI expression, the protein and messenger RNA (mRNA) levels were confirmed by Western blot analysis and RT-PCR. Protein (Figure 3B) and mRNA (Figure 3C) expression for the FcεRI α chain was inhibited by EDG and DC in a dose-dependent manner.

### 2.4. EDG and DC Effects on FcεRI-Mediated Activation of PTK, Syk, and Lyn

To investigate the suppressive effects of EDG and DC, protein tyrosine kinases (PTKs) such as Syk and Lyn were determined by Western blot analysis. pSyk and pLyn levels were elevated in CRA-1-stimulated cells, and EDG and DC treatments dose-dependently reduced the levels of these proteins compared with those in nonstimulated cells (Figure 4).

### 2.5. EDG and DC Effects on FcεRI-Mediated ERK ½ Activation

MAPK pathways are considered major mechanisms in IgE-mediated allergic reactions, and are involved in the regulation of other signaling factors. To determine whether the inhibitory effect of EDG and DC on the activation of basophils was mediated through MAPK pathways, we tested the effect of EDG and DC on CRA-1-induced MAPK phosphorylation. As shown in Figure 5, EDG and DC suppressed the CRA-1-induced phosphorylation of ERK 1/2.

### 2.6. EDG and DC Inhibited FcεRI-Mediated Calcium Influx and Degranulation

To examine the effects of these compounds on calcium influx and degranulation, intracellular calcium content [Ca^2+^]*i* and histamine release were measured with spectrofluorometric analysis, which was performed using specific probes Fura 2-AM and *o*-phthalaldehyde (OPA), respectively. The levels of [Ca^2+^]*i* (Figure 6A) and histamine release (Figure 6B) were dose-dependently inhibited by EDG and DC compared to nonstimulated cells.

### 2.7. EDG and DC Effects on FcεRI-Mediated Signaling Pathway in KU812F Cells

We examined the effect of EDG and DC on signaling pathways mediated by FcεRI, and detected the downregulation of FcεRI expression. Our results showed that FcεRI levels decreased, and that downregulation of FcεRI expression inhibited the phosphorylation of factors (PTKs such as Syk and Lyn and ERK 1/2) related to FcεRI-mediated downstream signaling (Figure 7). These results suggested that EDG and DC downregulated FcεRI-mediated signaling in human basophilic KU812F cells.

## 3. Discussion

In recent years, many natural products have been discovered as new functional foods and drugs, and prescribed as traditional herbal medicine to treat various disorders [21]. CZL has been traditionally used for treating various diseases, such as pneumonia, bronchitis, cough, the common cold, pharyngitis, bladder-related disorders, gastroenteric disorders, and hypertension [13,14,15,16,17,18,19]. EDG and DC are components of dried CZL; there have been no studies on their inhibitory effects in IgE-mediated allergic reactions (Figure 1). In the present study, the antiallergic effects of EDG and DC were investigated for the first time. To search for natural therapeutic products against allergic reactions, we isolated EDG and DC from CZL, and investigated their protective effects against allergic disorders in CRA-1-stimulated human basophilic KU812F cells under serumfree conditions. The cytotoxicity of EDG and DC on the proliferation of KU812F cells was examined to determine EDG and DC concentrations for all our subsequent studies. Our results indicated that these compounds exerted no toxic effects on the cells at concentrations of up to 100 μM (Figure 2).

FcεRI is a high-affinity IgE receptor expressed on the cell surface of mast cells and basophils [9,10,11]. Recently, much attention has been focused on searching for biologically active antiallergic compounds that downregulate FcεRI expression [12,22,23]. Our results showed that EDG and DC induced a decrease in protein, and that the mRNA levels of the cellular FcεRI α-chain may be associated with the suppression of cell-surface FcεRI expression (Figure 3).

Syk can induce the activation of mast cells and basophils, and their interaction with FcεRI and other signaling factors such as Lyn can regulate Syk and downstream signals [24,25]. Our results showed that EDG and DC inhibited the protein-expression levels of Syk and Lyn in CRA-1-stimulated cells (Figure 4).

MAPKs play an important role in the regulation of cellular signaling pathways, such as apoptosis, inflammation, and allergic-reaction pathways [26,27,28]. We found that EDG and DC downregulated ERK 1/2 activation in a dose-dependent manner (Figure 5). Our results showed that MAPK downregulation plays a crucial role in reducing EDG and DC-induced antiallergic action. It is likely that EDG and DC mediated the suppression of CRA-1-induced calcium influx and degranulation. Crosslinking resulted in the elevation of calcium influx and histamine release. EDG and DC inhibited the elevation of [Ca^2+^]*i* in CRA-1-stimulated KU812F cells (Figure 6A). Histamine is a potent inflammatory mediator, stored in secretory granules, and it is released in immunologically activated mast cells and basophils. Histamine in the medium is utilized as a marker of the degranulation of effector cells in IgE-mediated allergic reactions [29,30,31,32]. To examine the effects of EDG and DC on FcεRI-mediated degranulation, we assessed the levels of histamine released in CRA-1-stimulated KU812F cells, and determined its inhibition of EDG and DC (Figure 6B).

Taken together, the results of this study indicated, for the first time, that EDG and DC suppressed the activation of basophils by inducing degranulation following exposure to CRA-1. The inhibitory activities of EDG and DC were related to the downregulation of PTK and MAPK signaling. Further studies on the protective action of EDG and DC in anaphylaxis-induced animal models are necessary to confirm their potential therapeutic applications in the treatment of allergic diseases.

## 4. Materials and Methods

### 4.1. Plant Materials, Extraction, and Isolation

The whole CZL plant used in this study was purchased from Herbal Medicine Merchandise, Jecheon Hanbang Yakcho (www.jchanbang.com, Jecheoncity, Korea). A voucher specimen was deposited in the authors’ laboratory. Lyophilized whole-plant powder of CZL was extracted with MeOH at room temperature for 2 days. The extract was suspended in water and then fractionated sequentially with *n*-hexane, chloroform, and ethylacetate (EtOAc). EDG and DC were isolated from the EtOAc layer, which inhibited histamine release. The EtOAc layer was further fractionated with 70% MeOH using a Sephadex LH-20 (GE healthcare Bioscience, Uppsala, Sweden), which yielded Fractions I and II. The fractions were further fractionated on an HPLC system (Agilent 1100, Agilent, Santa Clara, CA, USA) with an octadecylsilane column (10 × 150 mm, Tosoh, Tokyo, Japan) at a flow rate of 3 mL/min by using a 5–100% aqueous MeOH gradient system (0.04% trifluoroacetic acid) as the mobile phase. EDG (6.0 mg) and DC (7.7 mg) were isolated from Fraction I at retention time of 22.6 and 19.8 min, respectively. These compounds were then subjected to electrospray ionization mass spectrometry (ESI-MS) (Agilent, Santa Clara, CA, USA), ^1^H-nuclear magnetic resonance (NMR), and ^13^C-NMR analysis.

*EDG*. yellowish powder; C_21_H_20_O_12_; ^1^H-NMR (600 MHz, CD_3_OD) δ 6.83 (1H, s, H-2′), 6.71 (1H, d, *J* = 8.3 Hz, H-6′), 6.68 (1H, d, *J* = 8.3 Hz, H-5′), 6.10 (1H, s, H-8), 6.08 (1H, s, H-6), 5.25 (1H, dd, *J* = 13.1, 2.5 Hz, H-2), 4.97 (1H, d, *J* = 6.2 Hz, H-1″), 3.93 (1H, br d, *J* = 7.6 Hz, H-5″), 3.50 (1H, br t, *J* = 8.3 Hz, H-4″), 3.40 (1H, overlapped with H-2″, H-3″), 3.38 (1H, overlapped with H-3″, H-2″), 3.03 (1H, dd, *J* = 16.5, 13.1, H-3a), 2.66 (1H, dd, *J* = 16.5, 2.5, H-3b). ^13^C-NMR (150 MHz, CD_3_OD) δ 197.2 (C-4), 170.7 (C-6″), 165.6 (C-8a), 163.6 (C-7), 163.3 (C-5), 145.6 (C-4′), 145.2 (C-3′), 131.7 (C-1′), 117.9 (C-6′), 114.9 (C-5′), 113.4 (C-2′), 103.6 (C-4a), 99.8 (C-1″), 96.7 (C-6), 95.6 (C-8), 79.4 (C-2), 76.2 (C-5″), 75.3 (C-3″), 73.1 (C-2″), 72.1 (C-4″), 42.8 (C-3); positive ESI-MS *m*/*z* 465.1 [M + H]^+^.

*DC*. whitish powder; C_9_H_6_O_4_, ^1^H-NMR (600 MHz, CD_3_OD) δ 7.87 (1H, d, *J* = 6.2 Hz, H-2), 6.24 (1H, d, *J* = 2.0 Hz, H-8), 6.11 (1H, d, *J* = 2.0 Hz, H-6), 6.09 (1H, d, *J* = 6.2 Hz, H-3). ^13^C-NMR (150 MHz, CD_3_OD) δ 182.0 (C-4), 164.8 (C-7), 162.1 (C-5), 158.6 (C-8a), 156.8 (C-2), 103.6 (C-4a), 98.9 (C-6), 93.6 (C-8); positive ESI-MS *m*/*z* 179.0 [M + H]^+^.

### 4.2. Cell Culture, Treatment, and Stimulation

The human basophilic KU812F cell line was obtained from the American Type Culture Collection and maintained in an RPMI-1640 medium (HyClone, Logan, UT, USA) supplemented with 10% heat-inactivated fetal bovine serum (FBS) (HyClone, Logan, UT, USA), antibiotics, and antimycotics (HyClone). Cells were cultured at 37 °C in a humidified atmosphere with 5% CO_2_, and passaged every 3–4 days. KU812F cells were treated with various concentrations of EDG and DC for 24 h under serumfree conditions, and stimulated with 10 μg/mL of CRA-1 (Kyokuto, Tokyo, Japan) in Tyrode buffer (137 mM NaCl, 2.7 mM KCl, 0.4 mM NaH_2_PO_4_, 1 mM MgCl_2_, 12 mM NaHCO_3_, 1.8 mM CaCl_2_) for the indicated times.

### 4.3. Cell-Viability Assay

The MTS assay was performed to investigate the cytotoxicity of EDG and DC. Cells (1 × 10^4^ cells/well) were incubated in a serumfree medium in the presence of EDG and DC at 37 °C. After 24 h of treatment, the MTS solution (Promega, Madison, WI, USA) was added to the fresh medium for 1 h, according to the manufacturer’s instructions. Then, absorbance at 490 nm was performed using a microplate reader (Molecular Device, Sunnyvale, CA, USA).

### 4.4. Flow Cytometric Analysis

The expression of FcεRI on the cell surface was measured using indirect immunofluorescence and flow cytometry (Epics^®^ XL^TM^, Beckman coulter, Brea, CA, USA). Briefly, pretreated KU812F cells (1 × 10^6^ cells) were incubated with CRA-1 (10 μg/mL) for 60 min on ice. Cells were then stained with fluorescein isothiocyanate (FITC)-conjugated F(ab’)_2_ goat antimouse IgG (20 μg/mL) (Jackson ImmunoResearch Lab., Baltimore, PO, USA) for 60 min on ice, washed with ice-cold phosphate-buffered saline, and then subjected to flow cytometry. Cells treated with the mouse IgG antibody (10 μg/mL) instead of CRA-1 were used as a negative control. The percentage of FcεRI-positive cells was calculated with an arbitrary cutoff position of 2%, as determined by the negative control. The percentage of cells expressing FcεRI on the cell surface is representative of three independent experiments.

### 4.5. Western Blot Analysis

The protein expression of nonstimulated and stimulated cells was measured by Western blotting analysis. Cells were treated with various concentrations of EDG and DC under serumfree conditions, nonstimulated for FcεRI expression, and stimulated with CRA-1 (10 μg/mL) for PTK and MAPK expression. Whole cell lysates were extracted with a cell lysis buffer containing 20 mM Tris-Cl (pH 8.0), 137 mM NaCl, 10% glycerol, 1% Triton X-100, 1 mM Na_3_VO_4_, 1 mM NaF, 2 mM EDTA, and a protease inhibitor cocktail (Roche, Penzberg, Germany). Equal amounts of protein were separated by 10% SDS-PAGE and transferred onto nitrocellulose membrane. The membrane was blocked with 5% skim milk in plain buffer (20 mM Tris pH 7.4 and 136 mM NaCl) at room temperature for 1 h, and incubated with primary antibodies overnight at 4 °C. The membrane was then incubated with 500 times diluted specific secondary horseradish peroxidase (HRP)-conjugated antibodies at room temperature for 1 h, and the immunoreactive bands were visualized using enhanced chemiluminescence detection reagents (Perkin Elmer, Waltham, MA, USA), in according to the manufacturer’s instructions.

### 4.6. Reverse Transcriptase Polymerase Chain Reaction (RT-PCR)

Total cellular RNA was isolated using TRIzol reagent (Gibco BRL, Gaithersburg, MD, USA) according to the manufacturer’s instructions. For complementary DNA (cDNA) synthesis, 1 μg of total RNA was reverse-transcribed using an oligo(dT)_20_ primer (Gibco BRL) and MMLV reverse transcriptase (Promega). The resultant cDNA samples were amplified by PCR using specific sense and antisense primers. Human glyceraldehyde-3-phosphate dehydrogenase (G3PDH) was used as control. The primer sequences used in this study were as follows: FcεRI α chain, sense 5′-CTTAGGATGTGGGTTCAGAAGT-3′ and antisense 5′-GACAGTGGAGAATACAAATGTCA-3′; G3PDH, sense 5′-GCTCAGACACCATGGGGAAGGT-3′ and antisense 5′-GTGGTGCAGGAGGCATTGCTGA-3′. PCR conditions for the FcεRI α-chain and G3PDH genes were as follows: denaturation at 94 °C for 30 s; annealing at 55 °C for 30 s; and extension at 72 °C for 1 min, with 18 cycles. Amplified PCR products were analyzed using 1% agarose gel electrophoresis and stained with ethidium bromide.

### 4.7. [Ca^2+^]i-Level Assay

The [Ca^2+^]*i* level was measured using the calcium-reactive fluorescence probe Fura 2-AM (Sigma, St. Louis, MO, USA). The pretreated KU812F cells were suspended in Tyrode solution and incubated with 2.0 μM of Fura 2-AM at 37 °C for 30 min. Next, the cells were washed three times, resuspended in fresh buffer, and stimulated with 10 μg/mL of CRA-1. Fura 2 fluorescence was monitored at excitation and emission wavelengths of 360 and 528 nm, respectively.

### 4.8. Histamine-Release Assay

Histamine level released from 10 μg/mL of CRA-1-stimulated KU812F cells was measured by a spectrophotometric assay [33]. Treated and stimulated cells were centrifuged, and supernatants were treated with 1N NaOH and 0.2% OPA (Sigma) for 40 min on ice. The reaction was terminated by the addition of 3N HCl. Fluorescence intensity was measured at excitation and emission wavelengths of 360 and 450 nm, respectively, using a microplate fluorescence reader (FLx800, BioTek, Winooskin, VT, USA).

### 4.9. Statistical Analysis

All measurements were conducted independently in at least triplicate. Data were expressed as mean ± SD. Significant differences between the control and the EDG and DC groups were determined by Student’s *t*-test at *p* < 0.05.

## Figures and Tables

**Figure 1 molecules-25-00994-f001:**
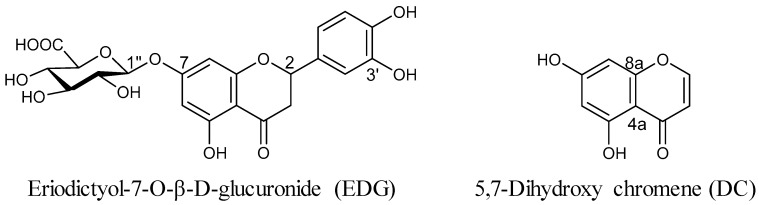
Chemical structures of (**left**) eriodictyol-7-*O*-*β*-d-glucuronide (EDG) and (**right**) 5,7-dihydroxy-4-chromene (DC) isolated from *Chrysanthemum zawadskii* var. *latilobum* (CZL).

**Figure 2 molecules-25-00994-f002:**
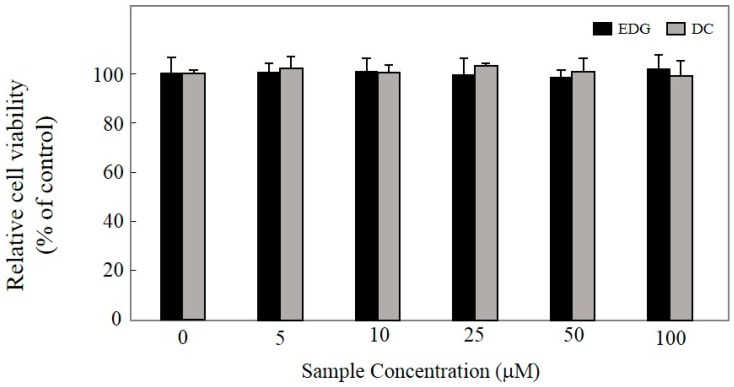
Cytotoxicity of EDG and DC. KU812F cells cultured in presence of different concentrations of EDG and DC for 24 h under serumfree conditions, and cell viabilities were determined via 3-(4,5-dimethylthiazol-2-yl)-5-(3-carboxymethoxyphenyl)-2-(4-sulfophenyl)-2*H*-tetrazolium (MTS) assay. Each determination as made in triplicate, and data expressed as mean ± SD.

**Figure 3 molecules-25-00994-f003:**
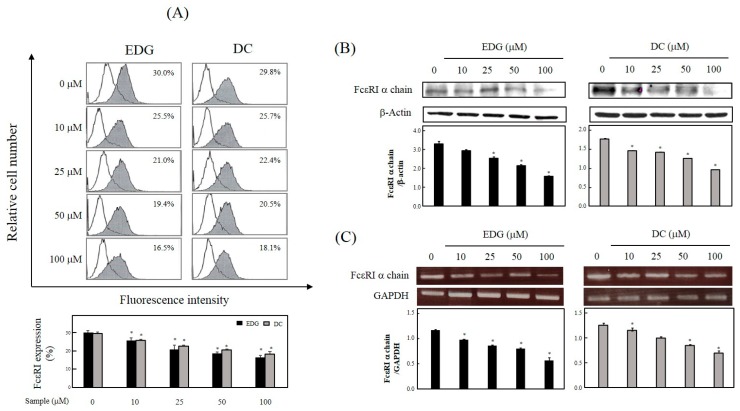
Effects of EDG and DC on FcεRI expression. KU812F cells were cultured in the presence of different EDG and DC concentrations (0, 10, 25, 50, and 100 μM) for 24 h under serumfree conditions. (**A**) Flow-cytometry analysis conducted using anti-FcεRI antibody (CRA-1) followed by staining with fluorescein isothiocyanate (FITC)-conjugated Fragment antigen binding, F(ab’)_2_ goat antimouse immunoglobulins. (**B**) Western blot analysis conducted using CRA-1 and β-actin. Protein amount in each band was quantified by densitometry. (**C**) Total RNA was prepared, and FcεRI α chain and internal control, glyceraldehyde 3-phosphate dehydrogenase (GAPDH) were detected by RT-PCR. Each value represents mean ± SD of three different experiments. Relative density calculated as ratio of each protein expression to β-actin and each mRNA level to GAPDH. * Values significantly different from control (* *p* < 0.05).

**Figure 4 molecules-25-00994-f004:**
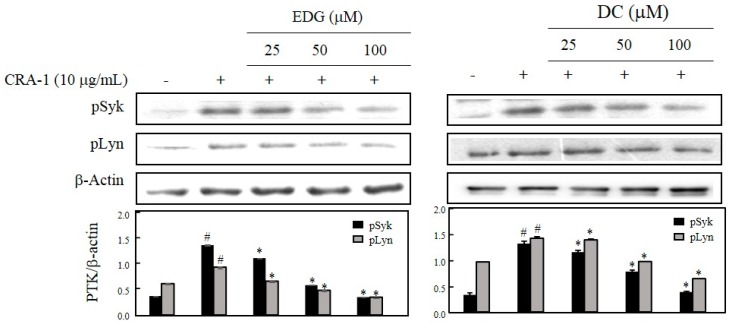
Effects of EDG and DC on FcεRI-mediated protein tyrosine kinases (PTKs) Syk and Lyn. Cells treated with various concentrations of EDG and DC under serumfree conditions and stimulated with CRA-1. Cellular lysates were obtained, and Syk, Lyn, and β-actin expression analyzed by Western blot analysis using corresponding antibodies. Relative density calculated as ratio of each protein expression to β-actin. # *p* < 0.05 vs. non-treated group; * *p* < 0.05 vs. CRA-1-treated group.

**Figure 5 molecules-25-00994-f005:**
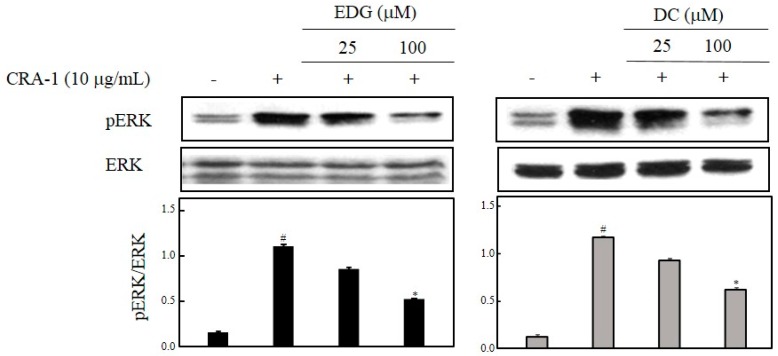
EDG and DC effects on FcεRI-mediated extracellular-regulated kinase (ERK) 1/2 activation. Cells treated with EDG and DC under serumfree conditions, and stimulated with CRA-1. Cellular lysate was obtained, and protein expression assessed via Western blot analysis using antiphospho ERK 1/2 and ERK 1/2 antibodies. Results presented as mean ± SD of three independent experiments. Relative density calculated as ratio of each protein expression to ERK. # *p* < 0.05 vs. non-treated group; * *p* < 0.05 vs. CRA-1-treated group.

**Figure 6 molecules-25-00994-f006:**
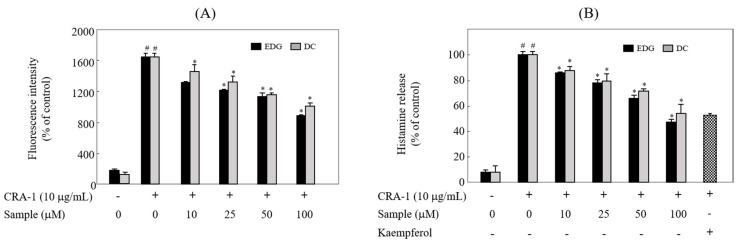
EDG and DC inhibited FcεRI-mediated calcium influx and degranulation. Cells treated with various concentrations of EDG and DC under serumfree conditions. (**A**) Cells incubated with Fura 2-AM and stimulated with CRA-1 to determine calcium influx. (**B**) To examine histamine content in the medium, cells were stimulated with CRA-1, and supernatants were treated with *o*-phthalaldehyde (OPA). These cells were determined via a spectrofluorometric method. Each value expressed as mean ± SD of three different experiments. # *p* < 0.05 vs. non-treated group; * *p* < 0.05 vs. CRA-1-treated group.

**Figure 7 molecules-25-00994-f007:**
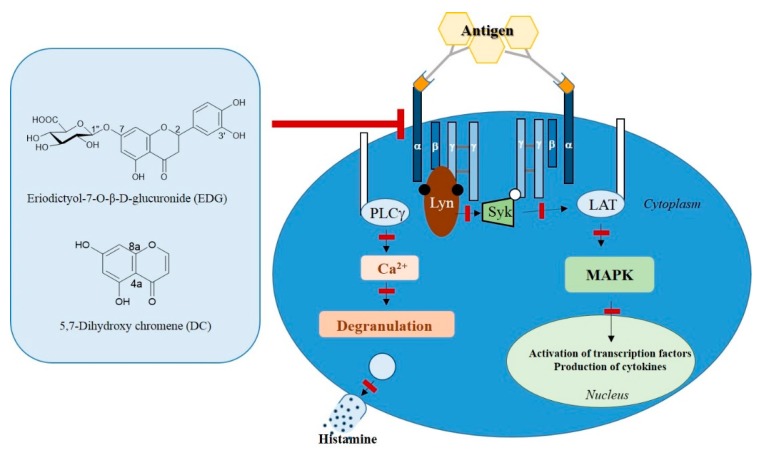
Effects of EDG and DC on FcεRI-mediated signaling pathway in KU812F cells.

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
