# Peer review of "Inhibitory Effects of Eriodictyol-7-O-β-d-glucuronide and 5,7-Dihydroxy-4-chromene Isolated from Chrysanthemum zawadskii var. latilobum in FcεRI-Mediated Human Basophilic KU812F Cell Activation"

_molecules, 2020, doi:10.3390/molecules25040994_

Round 1
Reviewer 1 Report
None
Author Response
We appreciate the time you have taken to review our manuscript and value your comments.
Reviewer 2 Report
The authors studied the EDG and DC, components of dried CZL, and their inhibitory effects on Ig E mediated allergic reaction in human basophilic cell lines. The design of experiments, to decipher in vitro molecular mechanisms is appropriate.
Minor comments:
Short-term culture (24h) is a concern. Longer culture would be more appropriate: it would answer the question: how long the effect of these compounds last, and if this eventually leads to cell apoptosis. Were the same Western blots re-probed for figures: 3b and 4? The pattern seems does not correspond between bands. Make sure images are not inverted and properly aligned. Control (Actin) expression is not shown in figure 5. The images need to be presented.Author Response
Comments and Suggestions for Authors
The authors studied the EDG and DC, components of dried CZL, and their inhibitory effects on Ig E mediated allergic reaction in human basophilic cell lines. The design of experiments, to decipher in vitro molecular mechanisms is appropriate.
Minor comments:
Short-term culture (24h) is a concern. Longer culture would be more appropriate: it would answer the question:
→ We thought that 24 hours culture is appropriate for signaling activation.
how long the effect of these compounds last, and if this eventually leads to cell apoptosis.
→ If these compounds induces apoptosis, it appear toxic within 24 hours.
Were the same Western blots re-probed for figures: 3b and 4? The pattern seems does not correspond between bands. Make sure images are not inverted and properly aligned. Control (Actin) expression is not shown in figure 5. The images need to be presented. Figure 3b and 4
→ Revised as request.
→ We appreciate the time you have taken to review our manuscript and value your comments.
Reviewer 3 Report
The present work entitle “Inhibitory Effects of Eriodictyol-7-O-ß-D-glucuronide and 5,7-Dihydroxy-4-chromene Isolated fromChrysanthemum zawadskii var. latilobum in FceRI-Mediated Human Basophilic KU812F Cells Activation” shows the antiallergenic effects of two related natural products. The ms is well written and shows a progressive studies from cytotoxicity to pathway activation.
Statistical analisys of data in fig 3 and 4 are confuse and not represent the data, * and # possible are missing.
Figure 7 shows pixeled must be improve
Line 64, author must indicated the reference used for NMR comparison
Line 72 indicated what meaning MTS?
Line 201, “EDG (6.0 mg) and DC (7.7 mg) were isolated from fraction I at retention time (RT) of 22.6 min.” Both compound have the same retention time? Must be corrected
Author Response
Comments and Suggestions for Authors
The present work entitle “Inhibitory Effects of Eriodictyol-7-O-ß-D-glucuronide and 5,7-Dihydroxy-4-chromene Isolated fromChrysanthemum zawadskii var. latilobum in FceRI-Mediated Human Basophilic KU812F Cells Activation” shows the antiallergenic effects of two related natural products. The ms is well written and shows a progressive studies from cytotoxicity to pathway activation.
Statistical analisys of data in fig 3 and 4 are confuse and not represent the data, * and # possible are missing.
→ Revised as request.
Figure 7 shows pixeled must be improve.
→ Revised as request.
Line 64, author must indicated the reference used for NMR comparison.
→ Added the reference as request.
Line 72 indicated what meaning MTS?
→ Revised MTS assay in Materials and Methods section.
Line 201, “EDG (6.0 mg) and DC (7.7 mg) were isolated from fraction I at retention time (RT) of 22.6 min.” Both compound have the same retention time? Must be corrected.
→ Revised as request.
→ We appreciate the time you have taken to review our manuscript and value your comments.
Reviewer 4 Report
Thank you for the opportunity to review your manuscript. Overall, I found the topic interesting, however, I believe that the paper requires a considerable amount of work before it is worthy of publication.
My main concern regarding the paper is that the entire results section is written extremely poorly as the authors restate the methods at the beginning of each section. Further, the authors do not clearly explain the results generated in the text, nor in the figure legends (Figures 2-6 inclusive). In fact, all figure legends contain rewritten methods, with no (or very little) explanation as to what exactly the results showed. It should be noted that it is not up to the reader to interpret the data, this is the responsibility of the authors. Also, the authors state that all experiments were conducted in at least triplicate. What is this supposed to mean?? As a scientist, I would like to know EXACTLY how many replicas you performed for each experiment. Please include as appropriate.
specifically:
line 61/62: delete sentence beginning EDG and ending chromatography. lines 68-72: delete all sentences (up to MTS assay). line 74: delete. As part of the results for this section, you need to explain them in slightly more detail whilst also including the concentrations tested. with this in mind, why did you only measure cytotoxicity up to 24 hours? I would advise a repeat of this section whilst also including an additional two points i.e. 48 and 72 hours to provide a more definitive indication of potential cytotoxic effects of both compounds. The extra time may show inducement of cytotoxicity as the experiment progresses. However, although this may not be the case, it is always prudent to be 100% certain. lines 82-85: delete all sentences (up to RT-PCR). Again, this is methods, not results. As per comments above, you need to explain your results more clearly. In figure 3A, you demonstrated significance, yet nowhere have you stated this in the text. You must be precise. lines 99-102: delete all sentences (up to CRA-1). Again, your figure 4 (and figures 5 and 6) shows significance but you have not included this result in your text under results. lines 127-130: delete all sentences (up toCRA-1). lines 131-133: delete sentence starting " The histamine content" and ending in "positive control."Other comments:
the following references are missing: line 34: sentence ending distribution line 35: sentence ending identified line 45: sentence ending reaction line 52: sentence ending hypertension line 54: sentence ending effects line 154: sentence ending disorders line 167: sentence ending expressionMinor comments:
line 35: add "the" between 'to' and 'Fc' line 40: delete "The" and capitalise Aggregation. Also, aggregation of what exactly???? line 41: delete "and" and replace with 'as' lines 47-52 is basically a repeat from elsewhere. Please rewrite accordingly line 121 and 241: capitalise Western for Western blot you use a lot of abbreviations, but in most instances, do not write them out in full in the first instance. Moreover, in some cases, you abbreviate (e.g. methanol/MeOH, written in full line 199, instead of being abbreviated), but then do not use it throughout the rest of your text. You must be precise. MeOH, CHCl3, EtOAc, EG (line 220), OPA (line 272) line 203 and 211, you write the compound names in full when you have already abbreviated them prior. Use the abbreviation or not, but please do not do both. in the cell viability section (from line 224), what temperature did you incubate at? Please include. line 247: you have stated RT but have not previously defined it as room temperature. Please amend. line 250: after ECL assay kit, please include name of manufacturer in brackets. Finally, I believe that your discussion could do with a bit more depth to it.Author Response
Comments and Suggestions for Authors
Thank you for the opportunity to review your manuscript. Overall, I found the topic interesting, however, I believe that the paper requires a considerable amount of work before it is worthy of publication.
My main concern regarding the paper is that the entire results section is written extremely poorly as the authors restate the methods at the beginning of each section. Further, the authors do not clearly explain the results generated in the text, nor in the figure legends (Figures 2-6 inclusive). In fact, all figure legends contain rewritten methods, with no (or very little) explanation as to what exactly the results showed. It should be noted that it is not up to the reader to interpret the data, this is the responsibility of the authors. Also, the authors state that all experiments were conducted in at least triplicate. What is this supposed to mean?? As a scientist, I would like to know EXACTLY how many replicas you performed for each experiment. Please include as appropriate.
→ We appreciate the time you have taken to review our manuscript and value your comments.
specifically:
line 61/62: delete sentence beginning EDG and ending chromatography.
→ Deleted as request.
lines 68-72: delete all sentences (up to MTS assay).
→ Deleted as request.
line 74: delete. As part of the results for this section, you need to explain them in slightly more detail whilst also including the concentrations tested. with this in mind, why did you only measure cytotoxicity up to 24 hours? I would advise a repeat of this section whilst also including an additional two points i.e. 48 and 72 hours to provide a more definitive indication of potential cytotoxic effects of both compounds. The extra time may show inducement of cytotoxicity as the experiment progresses. However, although this may not be the case, it is always prudent to be 100% certain.
→ We think that treatment of sample is sufficient for 24 h, because most signaling factors are activated within 24 h. If it should be thought of as apoptosis, cytotoxicity will appears within 24 h.
lines 82-85: delete all sentences (up to RT-PCR).
→ Deleted as request.
Again, this is methods, not results. As per comments above, you need to explain your results more clearly. In figure 3A, you demonstrated significance, yet nowhere have you stated this in the text. You must be precise.
→ Revised as request.
lines 99-102: delete all sentences (up to CRA-1).
→ Deleted as request.
Again, your figure 4 (and figures 5 and 6) shows significance but you have not included this result in your text under results.
→ Revised as request.
lines 127-130: delete all sentences (up toCRA-1).
→ Deleted as request.
lines 131-133: delete sentence starting " The histamine content" and ending in "positive control."
→ Deleted as request.
Other comments:
the following references are missing:
line 34: sentence ending distribution
line 35: sentence ending identified
line 45: sentence ending reaction
line 52: sentence ending hypertension
line 54: sentence ending effects
line 154: sentence ending disorders
line 167: sentence ending expression
→ Added the reference as request.
Minor comments:
line 35: add "the" between 'to' and 'Fc' line 40: delete "The" and capitalise Aggregation.
→ Revised as request.
Also, aggregation of what exactly????
→ Aggregation means crosslinking of FceRI molecules attached to allergen-specific IgE antibody.
line 41: delete "and" and replace with 'as'
→ Revised as request.
lines 47-52 is basically a repeat from elsewhere. Please rewrite accordingly
→ Revised as request.
line 121 and 241: capitalise Western for Western blot you use a lot of abbreviations, but in most instances, do not write them out in full in the first instance.
→ Revised as request.
Moreover, in some cases, you abbreviate (e.g. methanol/MeOH, written in full line 199, instead of being abbreviated), but then do not use it throughout the rest of your text. You must be precise. MeOH, CHCl3, EtOAc, EG (line 220), OPA (line 272) line 203 and 211, you write the compound names in full when you have already abbreviated them prior.
→ Revised as request.
Use the abbreviation or not, but please do not do both. in the cell viability section (from line 224), what temperature did you incubate at?
→ Added the incubation temperature as request.
Please include. line 247: you have stated RT but have not previously defined it as room temperature.
→ Revised as request.
Please amend. line 250: after ECL assay kit, please include name of manufacturer in brackets.
→ Revised as request.
Finally, I believe that your discussion could do with a bit more depth to it.
→ Thank you very much for your kind review.